# Identification of Endometriosis Pathophysiologic-Related Genes Based on Meta-Analysis and Bayesian Approach

**DOI:** 10.3390/ijms26010424

**Published:** 2025-01-06

**Authors:** Jieun Kang, Kwangjin Ahn, Jiyeon Oh, Taesic Lee, Sangwon Hwang, Young Uh, Seong Jin Choi

**Affiliations:** 1Department of Obstetrics and Gynecology, Yonsei University Wonju College of Medicine, 20, Ilsan-ro, Wonju-si 26426, Republic of Korea; k131730@hanmail.net; 2Department of Laboratory Medicine, Yonsei University Wonju College of Medicine, 20, Ilsan-ro, Wonju-si 26426, Republic of Korea; kjahn123@yonsei.ac.kr; 3Department of Global Medical Science, Yonsei University Wonju College of Medicine, 20, Ilsan-ro, Wonju-si 26426, Republic of Korea; jiyeon06@yonsei.ac.kr; 4Department of Family Medicine, Yonsei University Wonju College of Medicine, 20, Ilsan-ro, Wonju-si 26426, Republic of Korea; ddasic123@yonsei.ac.kr; 5Department of Precision Medicine, Yonsei University Wonju College of Medicine, 20, Ilsan-ro, Wonju-si 26426, Republic of Korea; arsenal@yonsei.ac.kr

**Keywords:** endometriosis, differential expression analysis, Bayesian analysis, network analysis

## Abstract

Endometriosis is a complex disease with diverse etiologies, including hormonal, immunological, and environmental factors; however, its exact pathogenesis remains unknown. While surgical approaches are the diagnostic and therapeutic gold standard, identifying endometriosis-associated genes is a crucial first step. Five endometriosis-related gene expression studies were selected from the available datasets. Approximately, 14,167 genes common to these 5 datasets were analyzed for differential expression. Meta-analyses utilized fold-change values and standard errors obtained from each analysis, with the binomial and continuous datasets contributing to endometriosis presence and endometriosis severity meta-analysis, respectively. Approximately 160 genes showed significant results in both meta-analyses. For Bayesian analysis, endometriosis-related single nucleotide polymorphisms (SNPs), the human transcription factor catalog, uterine SNP-related gene expression, disease–gene databases, and interactome databases were utilized. Twenty-four genes, present in at least three or more databases, were identified. Network analysis based on Pearson’s correlation coefficients revealed the *HLA-DQB1* gene with both a high score in the Bayesian analysis and a central position in the network. Although *ZNF24* had a lower score, it occupied a central position in the network, followed by other *ZNF* family members. Bayesian analysis identified genes with high confidence that could support discovering key diagnostic biomarkers and therapeutic targets for endometriosis.

## 1. Introduction

Endometriosis is an enigmatic disease characterized by the presence of endometrial glands and stroma outside the uterine endometrium, inhabiting diverse locations, including the ovaries, peritoneum, and extraperitoneal organs [1]. Endometriosis, characterized by a prevalence of as high as 10% among women of reproductive age, gradually progresses and evolves into a chronic inflammatory condition. Medical treatments for endometriosis, such as gonadotropin-releasing hormone agonists, dienogest, and oral contraceptives, along with surgical treatments often cause a reduction in functional ovarian tissue and ovarian reserve, making pregnancy challenging for both natural conception and assisted reproductive techniques [1,2].

Endometriosis is recognized as a complex condition influenced by hormonal, immunological, and environmental factors [1]. In 1927, John A. Sampson proposed that retrograde menstruation allowed endometrial tissue to enter the peritoneal cavity, adhere, and invade the epithelium, with inadequate immune responses enabling its persistence and growth [1,3,4]. Another theory suggested that coelomic epithelium metaplasia, driven by environmental factors, played a key role in its development. Although retrograde menstruation occurred universally in reproductive aged women, only about 10% developed endometriosis, indicating the involvement of additional factors such as altered immunity, metastasis, genetic predispositions, environmental triggers, and a complex interplay of gene–environment interactions [1,4,5].

Given the exceedingly intricate pathophysiology of endometriosis, researchers have yet to establish definitive biomarkers for diagnosis [5]. Although laparoscopic surgery was traditionally considered the gold standard for diagnosing endometriosis [6,7], it is no longer routinely performed for diagnostic purposes. With medical treatment, independent of surgical intervention, constituting a substantial portion of therapeutic approaches, the need for non-invasive diagnostic methods distinct from historical approaches has increased; however, they require high specificity [2,6]. Similarly to other complex diseases of multifactorial origin, the identification of genes associated with high-penetrance endometriosis remains elusive. Genetic exploration holds promise for the identification of critical pathophysiological pathways [2,6].

In this study, we analyzed gene expression data related to endometriosis from open databases and identified genes associated with the presence and severity of the disease by combining multiple genome-wide endometriosis-related transcriptomes (Figure 1). However, there are crucial challenges of gene-level studies, such as heterogeneity among datasets, high cost, and relatively small dataset size, which could provide false positive or negative findings [8,9]. We implemented meta-analysis [10], a traditional method to successfully deal with the genetic study’s challenges, to identify reliable endometriosis-related biomarkers (Figure 1). Recently, a Bayesian approach, which integrates computational results with known knowledge (database and published papers), was used to screen key genes involving disease pathophysiology [11,12,13,14]. We utilized the Bayes method, also called convergence functional genomics, to narrow down the hub genes involving the pathophysiology of endometriosis.

## 2. Results

### 2.1. Data Exploration and Selection

Obstetrics and gynecology specialists selected data from the Gene Expression Omnibus (GEO) for the meta-analysis. GSE6364 was used to analyze gene expression in endometrial tissue from patients with moderate-to-severe endometriosis and a control group. Similarly, GSE73622 examined gene expression in endometrial tissue samples from a patient–control study in which researchers reprogrammed collected cells into endometrial mesenchymal stem cells (MSCs) and subsequently differentiated them into endometrial stromal fibroblasts (SFs) using progesterone. GSE141549 explored gene expression in endometrial and peritoneal tissues from patients with endometriosis and healthy women as well as in various lesions within the peritoneal cavities of patients. Uniquely, data from 115 patients were analyzed on a separate platform (GPL10558), while data from 53 additional controls were analyzed on the GPL13376 platform. Analysts excluded data from the control group in GSE141549-GPL13376 if medication usage or menstrual cycle details were not recorded.

### 2.2. Indentification of Differentially Expressed Genes

A laboratory medicine specialist conducted gene expression analyses across all five GSE datasets. For handling gene expression data, R (R Foundation for Statistical Computing, Vienna, Austria) was used. Given that GSE6364 contains data with gene expression values exceeding 8000, the analysis utilized log-transformed values. Each GSE sample was trimmed using normalizeQuantile function. Trimmed data were subjected to principal component analysis (PCA) to identify any batch effects, and any identified batches were designated as confounders during the limma analysis [20].

GSE141549-GPL10558 and GSE141549-GPL13376 include the disease grade results for each patient. These two datasets can be used to analyze disease progression. To analyze the genes associated with endometriosis, we adjusted for GSE141549-GPL13376 and generated a binomial distribution for the patient–control groups instead of using continuous variables for disease grades. Consequently, the GSE6364, GSE73622-MSC, GSE73622-SF, and GSE141549-GPL13376 datasets acquired binomial distributions, allowing us to calculate fold-change (FC) values for the presence of endometriosis. Enrichment analyses were performed using Gene Ontology [21] and the Kyoto Encyclopedia of Genes and Genomes [22]. When common genes between differentially expressed genes (DEGs) and genes within the enrichment pathway were revealed, the false discovery rate (FDR) was calculated. A variety of pathways, including lipid metabolism, protein degradation, cell metabolism, and mRNA signaling (Appendix A), were identified in each GSE dataset.

Figure 2 compares the FC distribution across the five GSE datasets and the FC correlation for the same genes identified in the two GSE datasets. A positive FC correlation was observed only between the GSE73622-MSC and GSE73622-SF datasets, which used the same platform as the experimental group. However, GSE141549-GPL13376 and GSE141549-GPL10558, although both were sourced from endometriosis tissues of the same patients and used different platforms, showed a negative correlation in FC for the same observed genes. This difference, attributed to GSE141549-GPL13376 focusing on presence and GSE141549-GPL10558 focusing on severity, led us to separate the meta-analyses for presence and severity.

### 2.3. Meta-Analysis on DEGs

The results of the meta-analysis for presence, which are based on binomial groups, cannot be directly used for the severity meta-analysis. However, by transforming the controls and patients into continuous groups with values of 0 and 1, respectively, they were included in the severity analysis. Additionally, the GSEs used for the presence analysis were indirectly incorporated into the severity analysis. Each gene had log fold-change (logFC) and standard error (SE) data for each GSE, which were used to perform the meta-analysis using METAL based on the inverse variance-weighted average method (IVW) [10]. Analysis using METAL provided the z-scores and *p* for each gene. DEGs with *p* less than 0.05 were related to cellular responses and protein degradation through pathway enrichment analysis (Appendix A). Genes that were significant (*p* < 0.05) had a z-score absolute value greater than 1.96. Approximately, 160 genes had a z-score of less than 1.96 in both meta-analyses (Figure 3).

### 2.4. Bayesian Analysis of Endometriosis Severity-Related Genes

For the 160 genes filtered by the *p* obtained from the meta-analyses results, Bayesian analysis was applied to identify endometriosis-related upstream genes. The scoring matrix consisted of the following five types of prior knowledges: genome-wide association study (GWAS) for SNPs associated with the development of endometriosis [15]; human transcription factors (TFs) catalog the affect transcription of pathological genes [16]; SNPs associated with gene expression (eSNP), known as expression quantitative trait loci (eQTL), that affect the amount of pathological protein expression [17]; DigSee from disease–gene database [18]; and interactome database containing protein–protein interaction (PPI) of proteins made from pathological genes with other proteins [19]. The genes were scored based on the number of datasets in which they were included, and the confirmed genes with higher scores were selected as priority genes.

The final list contained 24 genes, with the highest priority being the *PPARA* and *HLA-DQB1* genes colored in purple. Magenta was selected for three Bayesian datasets, including *EP300*, *MAP2K6*, *ZC3HAV1*, *EIF2S1*, *VIM*, *ZNF436*, and *MRRF*, and the green color was set for the next ranking including *SETBP1*, *CEP152*, *TSPAN14*, *GNG5*, *BRD4*, *RPS11*, *GDI2*, *MAPK7*, *TXN*, *UTP15*, *ZNF134*, *ZNF304*, *ZNF786*, *ZNF24*, and *ZNF550* (Figure 4). Higher levels indicated a stronger association with the pathogenesis of endometriosis.

### 2.5. Network Construction by Correlation Analysis

Pearson correlation analysis of the 24 genes was conducted based on the t-values identified from the 6 GSE datasets, and 21 gene pairs with *p* below 0.05 were identified (Figure 5b). This analysis revealed a major cluster consisting of 19 genes and several individual genes surrounding these clusters (Figure 5c). The major cluster was centered on *HLA-DQB1* and *ZNF24*. In particular, *HLA-DQB1* scored high in the Bayesian analysis (Figure 4). *PPARA*, which received the highest score in the Bayesian analysis, was depicted as the surrounding gene paired with *ZNF134* and *UTP15*. The *ZNF* family, which includes highly upstream genes (Figure 5a), lies together, whereas *ZNF24* is mainly centered on *HLA-DQB1*.

## 3. Discussion

The Bayesian method includes two main parts: the optimization of likelihood and verification by prior knowledge. We arranged the results of the meta-analysis as the likelihood part and multiple lines of evidence, including GWAS [15], TF [16], eQTL [17], and the disease–gene database [18], as the prior part. Different datasets derived DEG results using samples collected from entirely distinct individuals, leading to varying coefficient values for the same gene. However, this heterogeneity was utilized as a hyperparameter to structure each dataset into models of identical format. By integrating these models, the meta-analysis addressed uncertainties observed across raw datasets by optimizing the likelihood [23]. Furthermore, leveraging datasets obtained from actual experiments, rather than entirely virtual foundations, enabled the rapid construction of models based on prior knowledge [24]. This approach provided robust evidence for building models aimed at validating research hypotheses. Bayesian’s flexible nature allows it to be incorporated into meta-analysis-based studies and helps reduce the time and cost of biomarker discovery.

The Bayesian approach could reduce false positive results by using prior pieces of knowledge that are sufficiently reliable without special statistical calculations [13,14]. Additionally, integrating DEG analysis results from various datasets through meta-analysis involved iterative modeling, which, as a combination of prior and likelihood, delivered more reliable outcomes compared to simple statistical analyses [24]. Furthermore, validating the selected genes using credible genetic network databases enhanced the quality of pathophysiological identification. By adopting sufficiently well-founded signals, the Bayesian methods can show another way of looking at results beyond “*p*-value hysteria and fantasy” [25].

The treatment of endometriosis remains challenging even with the use of specific combinations of medical, surgical, and psychological approaches. Infiltration into various organs, including the uterus, ovaries, and pelvic reproductive organs as well as the rectosigmoid colon, bowel, peritoneum, and diaphragm, complicates management and contributes to high recurrence rates. Meta-analyses and systematic reviews have shown that pain recurs in 20.5% of cases at three years and 43.5% at five years, whereas the recurrence of lesions larger than 10 mm is reported in 9% and 28% of cases, respectively [7]. Severe forms of endometriosis, such as deep infiltrative endometriosis, cause chronic pelvic pain, dysmenorrhea, and dyspareunia, significantly reducing quality of life. It also leads to infertility, which affects women across multiple life stages [5]. As a chronic inflammatory disease with high recurrence rates, severe forms of endometriosis, including deep and extra-abdominal endometriosis, require new therapeutic approaches beyond the traditional medicosurgical treatments. Our analysis provides a novel perspective on the severe forms of endometriosis that continue to recur despite existing treatments and offers an alternative understanding of this condition.

The comparative analysis of gene expression profiles between pathological and healthy tissues is a robust methodology that provides valuable insights into the underlying cellular processes implicated in the etiology of various diseases. This approach serves as the cornerstone for comprehending the molecular intricacies that drive pathological conditions [26]. Previous studies have analyzed the changes in gene expression associated with endometriosis using various microarray platforms. Some studies have compared gene expression between endometriotic lesions and normal endometrial tissues [27,28]. In certain studies, comparisons were made between gene expression in patients with endometriosis and healthy controls [29,30].

*HLA-DQB1*, also known as major histocompatibility complex class II DQ beta 1, is located on the short arm of chromosome 6 at position 21.31. Fagerberg et al. demonstrated that *HLA-DQB1* was expressed at low levels in all tissues, particularly in the lungs, lymph nodes, and spleen [31]. Typically, specific genotypes of HLA-DQB1 were studied with respect to the development of autoimmune diseases [32,33]. However, a recent study by Xu et al. revealed that increased methylation of *HLA-DQB1* is associated with rheumatoid arthritis [34]. In our study, the Bayesian analysis was conducted following differential expression analysis, allowing for the identification of significant genes without determining whether they were upstream or downstream. This approach has yielded important insights, suggesting that endometriosis may also be classified as an autoimmune disease, indicating a significant achievement in understanding its pathogenesis.

*PPARA*, referred to as peroxisome proliferator-activated receptor alpha, is located on the long arm of chromosome 22 at position 13.31. Fagerberg et al. revealed that *PPARA* is weakly expressed in all organs, but high levels are observed in the kidney, heart, and small intestine [31]. The activation of *PPARA* is well known to be associated with lipid metabolism [35], and lipid agonists are being explored for their therapeutic effects in various diseases based on this mechanism [36]. *PPARA* scored highly in the Bayesian analysis as a gene linked to obesity. In this study, *PPARA* was identified to be downstream of endometriosis. However, given that lipid metabolism was not strongly associated with endometriosis, the connection between *PPARA* and endometriosis is thought to be more related to angiogenesis in endometriotic lesions, as recently reported by Pergialiotis et al. [37], rather than lipid metabolism.

PPARα activates genes required for fatty acid β-oxidation and represses IL-6 gene expression by regulating the AP-1 and NF-kB pathway [38]. HLA-DQB1 is expressed on antigen-presenting cells, which present exogenously derived peptides to CD4+ T cells to elicit an immune response. The etiological link between HLA-DQB1 and endometriosis was found at the DNA level [39]. From the results of this study, it remained unclear exactly how and to what extent PPARA and HLA-DQB1 contributed to the presence and severity of endometriosis. However, based on their association with pathways involved in immune response regulation, it was suggested that endometriosis was linked to an abnormal immune state. Given the intricately intertwined signaling pathways of the immune response, it was likely that the precise pathophysiology of endometriosis had not yet been fully elucidated. The findings of this study may have served as a guiding light in navigating this uncertain landscape.

The zinc finger protein family was linked, either individually or collectively, to a variety of diseases, including adenocarcinoma, squamous cell carcinoma, and Parkinson’s disease [40,41,42,43], highlighting the lack of a clearly defined pathogenic mechanism. However, in this study, several *ZNF* family members were identified upstream and formed part of the main cluster in the network analysis. *ZNF436* was selected in meta-analysis and ranked high in the Bayesian approach (Figure 4). ZNF436 was suggested to function as a negative regulator of mitogen-activated protein kinases (MAPKs), in which the signaling pathway regulated cell proliferation [44,45]. The ectopic expression of ZNF304 in primary CD4+ cells silences human immunodeficiency viral transcription [46]. The upstream of the *ZNF* family was hypothesized to contribute to immune dysregulation and an inadequate immune response, potentially leading to the onset and progression of endometriosis.

## 4. Materials and Methods

### 4.1. Dataset Selection Criteria

The selection of appropriate gene expression datasets from the GEO involved narrowing down results through several steps. The initial search term was “endometriosis,” and studies focusing on Homo sapiens were retained. To analyze the entire human genome, the study type was specified as “Expression profiling by array”. Datasets containing sufficient sample sizes with both patient and normal control groups were reviewed. Comparative studies evaluating responses to drugs were excluded to focus on genes associated with the pathophysiological characteristics of the disease. GSE6364 was identified as a dataset containing 21 patient samples and 16 control samples, providing gene profiling data of the endometrium. GSE73622 included 22 patient samples and 28 controls, analyzing the gene expression of dedifferentiated cells from the endometrium in endometriosis. GSE141549, with 335 patient samples and 73 controls, featured the largest sample size and provided transcriptome analysis across various stages of endometriosis and multiple tissue types.

### 4.2. Pre-Processing

The expression values of the downloaded GSE datasets were adjusted to ensure that the maximum and minimum values were within a consistent range. When the maximum values were excessively large, log transformation was applied. Because log transformation cannot be performed for negative values, an appropriate constant was added to all values to ensure that the minimum value was greater than zero before the transformation. The data were normalized to ensure that each sample exhibited a consistent distribution of expression values [20]. The log transformation used for normalization employed natural logarithms. Since the distribution of gene expression values varied across samples, errors could arise during analysis. To address this, normalizeQuantiles function from the limma package was applied to complete the normalization process [20]. PCA was conducted to examine the clustering patterns of the samples and assess the relationship between various factors included in the sample information. Factors showing strong correlations were identified as potential confounders.

### 4.3. Differential Expression Analysis

After reviewing the sample information, patient and control groups were designated. For differential expression analysis, factors identified as confounders during pre-processing were incorporated into the model design, and the limma package was used for analysis [20]. Variables identified as confounders through PCA were handled as factors using the as.factor function within R’s model.matrix, effectively mitigating batch effects. This design enabled the use of the lmFit function to construct linear models for continuous variables in the samples. The resulting linear models were analyzed statistically using the eBayes function. The topTable function organized the statistical outcomes into a data frame, facilitating the selection of genes. The results provided FC values between the patient and control groups for each gene, presented as log values. Additionally, *p* and t-values were analyzed. As the product of the t-value and SE equals logFC, the SE was manually calculated. *p* and logFC were used to generate volcano plots.

### 4.4. Meta-Analysis

The GSE6364, GSE73622-MSC, and GSE73622-SF datasets provide gene expression values for both healthy individuals and patients, allowing for binomial differential expression analysis. GSE141549-GPL13376 also included both healthy individuals and patients with the disease stage recorded, enabling both binomial and continuous differential expression analysis. GSE141549-GPL10558 included only patients, allowing for continuous differential expression analysis based on the disease stage. Meta-analysis of the presence of endometriosis was performed using four GSE datasets suitable for binomial analysis. For endometriosis severity, a meta-analysis was conducted by integrating two GSE datasets suitable for continuous differential expression analysis with the results of the endometriosis presence meta-analysis. Because the present meta-analysis results pertained to the control and patient groups, they were reassigned as 0 and 1, respectively, and then transformed into a continuous scale for meta-analysis. A meta-analysis was conducted using previously obtained logFC and SE values from the METAL program [10]. *p* and z-scores were calculated for each gene.

### 4.5. Bayesian Analysis

Bayesian analysis was used to identify the upstream genes associated with endometriosis through evidence collection. The five databases used as evidence included GWAS, TFs, eQTL, DigSee, and PPIs.

The statistical summary related to endometriosis obtained from the GWAS catalog contained information regarding genetic variation by calculating the relationship between SNPs and disease occurrence using logistic regression [15]. This database identifies SNP-trait associations through literature searches, and then extracts reported traits, significant SNP-trait associations, and sample metadata.

The TF catalog was derived from the study by Lambert et al. [16], who identified human transcription factors and their functional characterization based on the important role of TFs as master regulators of chromatin and transcription. This database contains 2765 genes for which proteins were examined manually by assembling lists of inferred TFs from multiple sources, and 1639 genes were identified as actual TFs based on validated experiments.

Genomic variants in uterine tissues extracted using the gene expression quantification method at the transcriptomic level were retrieved from the eQTL catalog [17]. This database provides statistical results for cis-eQTLs, which indicate the association between gene expression and SNPs, calculated by linear regression.

DigSee is a search engine that applies text-mining methods to MEDLINE abstracts for cancer-related research on gene–disease interactions [18]. By entering the keyword endometriosis in the search engine, 847 disease-related genes were identified.

The PPI network information was acquired from the STRING database [19]. STRING is a large database and search engine that collects materials on functional associations from publicly accessible sources, such as individual high-throughput studies. Seven lines of evidence support STRING’s use of public data as an information base: three prediction streams based on genomic contextual information, and one each for co-expression, text mining methods, experiment-based data, and previously curated pathway and protein complex knowledge.

### 4.6. Correlation and Network Analysis

Pearson’s correlation analysis was conducted on the 24 upstream genes identified in the Bayesian analysis, using t-values from the 6 differential expression analyses (Figure 2) and z-scores from the 2 meta-analyses (Figure 3a,b). In the correlation analysis, only the top 5% and bottom 5% of the derived degrees were used to form the edges of each node, which were then applied to the network analysis.

## 5. Conclusions

Selecting specific genes and validating their pathophysiological roles in analyzing gene–disease associations requires extensive time and labor [47]. This study integrated the results from multiple gene expression studies and conducted differential analyses of gene presence in patient–control comparisons and severity associations in stage analysis. This approach systematically narrowed down 160 genes from over 20,000 based on evidence. Furthermore, Bayesian analysis was employed to identify upstream genes. The regulatory genes identified through this process demonstrated substantial statistical significance from the perspective of data analytics. Although experimental validation of the roles of these genes in disease development is still required, this study established a strong foundation by identifying key genes for future research, making a significant contribution to the field.

## Figures and Tables

**Figure 1 ijms-26-00424-f001:**
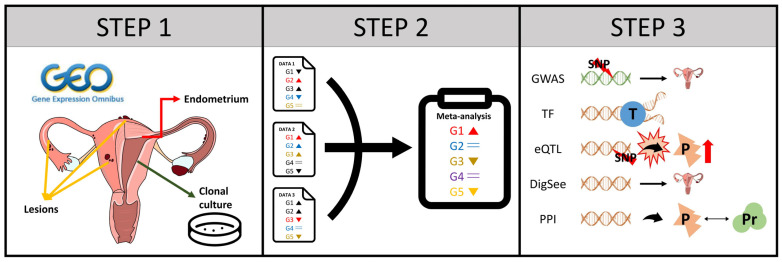
A scheme for the identification of genes associated with endometriosis. (STEP 1) A multi-disciplinary team including obstetrics and gynecology doctors, bioinformatics experts, and laboratory medicine doctors conducted literature and database reviews to select candidate transcriptome datasets related to endometriosis. The GSE6364 dataset analyzed gene expression in endometrial tissues from control groups and patients with moderate-to-severe endometriosis (indicated by red arrows). Similarly, GSE73622 differentiated endometrial mesenchymal stem cells and endometrial stromal fibroblasts from the endometrial tissue using progesterone were utilized to analyze gene expression (indicated by green arrows). GSE141549 examined endometrial and peritoneal tissues as well as various lesions in the peritoneal cavity of patients with endometriosis and healthy women (indicated by yellow arrows). (STEP 2) The algorithm conducts a differential expression analysis on datasets from the three GSE studies. A meta-analysis using the inverse variance-weighted method identified genes that were ultimately associated with endometriosis. (STEP 3) The Bayes method is implemented to identify the endometriosis upstream genes using five lines of external evidences: genome-wide association studies (GWAS) on single nucleotide polymorphism (SNP) associated with disease occurrence [15]; transcription factors (TF) affecting the transcription (T) of pathological genes [16], which identifies genes related to pathophysiology; expression quantitative trait loci (eQTL) reflecting the expression of SNP related to pathological proteins [17]; disease–gene database, known as DigSee [18]; and protein–protein interaction (PPI) data showing interactions between pathological proteins (P) and other proteins (Pr) [19].

**Figure 2 ijms-26-00424-f002:**
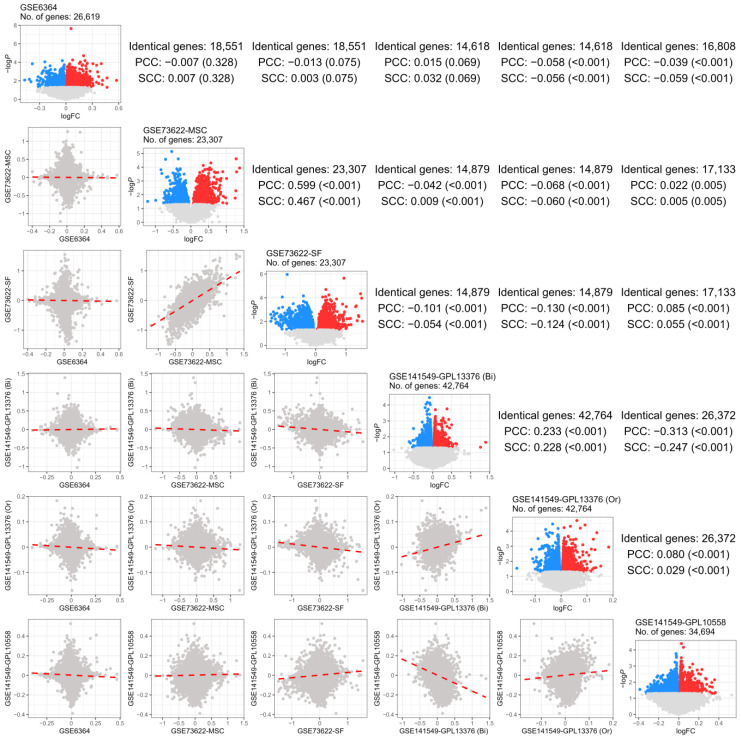
A comparison of global endometriosis-related signatures among five transcriptomic datasets. The volcano plots placed in the diagonal panels indicate cohort-specific endometriosis-related signatures (FC between two conditions or among three statuses). The specified order of GSE datasets, GSE6364, GSE73622-MSC, GSE73622-SF, GSE141549-GPL13376, and GSE141549-GPL10558 was followed, with volcano plots from differential analysis for each dataset arranged diagonally in the center. In the volcano plots, dots represent genes, with those having a *p* of 0.05 or higher shown in pale gray. Among the significant genes, red indicates upregulated genes, while blue means downregulated genes. Values placed in the upper triangle panels indicate the degree of correlation matched with the lower triangle plots evaluated by PCC and SCC. The *p* for each correlation coefficient appears in parentheses. A gray dot of correlational plots located in the lower triangle panels denotes a single identical gene contained in both gene expression datasets, and the matched red dashed line describes the correlation between two dataset obtained from linear regression. Abbreviations: FC, fold-change; MSC, mesenchymal stem cell; SF, stromal fibroblast; PCC, Pearson’s correlation coefficient; SCC, Spearman’s correlation coefficient; Bi, binomial; Or, ordinal.

**Figure 3 ijms-26-00424-f003:**
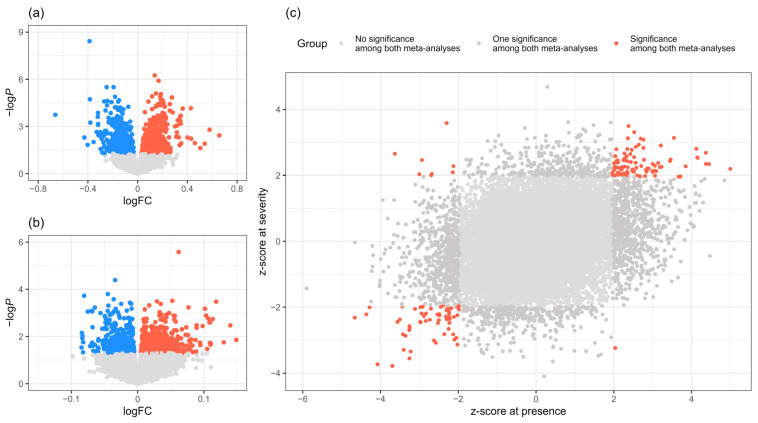
The identification of the endometriosis-related genes based on meta-analysis. (**a**) The IVW-based meta-analysis combined the four lists of global FC values between the presence and absence of endometriosis of four gene expression datasets (GSE6364, GSE73622-MSC, GSE73622-SF, and GSE141549-GPL13376). (**b**) The IVW combined the two lists of endometriosis severity-related signatures (GSE141549-GPL13376 and GSE141549-GPL10558) with the meta-analyzed FC values related to endometriosis presence (**a**). (**c**) The scatter plot indicates the comparison of meta-analyzed z-scores between presence and severity. A point represents an individual gene, and red points (about 160 genes) denote cases having 1.96 or more z-scores in both binomial (*x*-axis) and step-wise (*y*-axis) statuses. Abbreviations: IVW, inverse variance-weighted average method; FC, fold-change.

**Figure 4 ijms-26-00424-f004:**
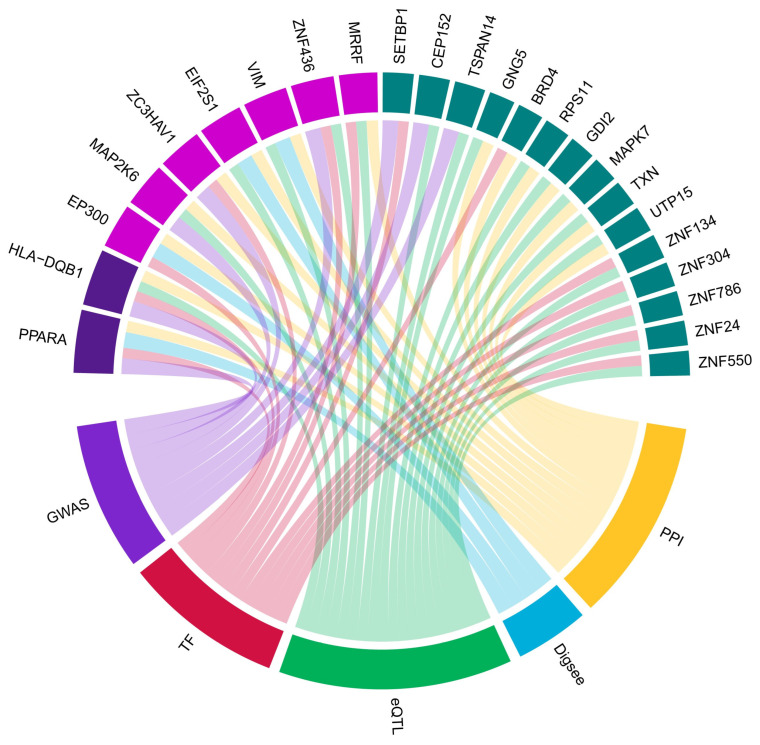
The identification of regulators related to endometriosis based on the Bayesian approach. Cells placed on the upper panel indicate candidate regulators. The colors of each cell are determined based on the strength of external evidence. Cells on the lower panel indicate the prior knowledge section of the Bayesian approach. The prior knowledge includes multiple lines of external evidence, including endometriosis-related SNPs (GWAS) [15], the human TF catalog [16], uterus eSNP (eQTL) [17], the disease–gene database (DigSee) [18], and the interactome database [19]. Abbreviations: SNP, single nucleotide polymorphism; GWAS, genome-wide association study; TF, transcription factor; eSNP, SNP-associated gene expression; eQTL, expression quantitative trait loci.

**Figure 5 ijms-26-00424-f005:**
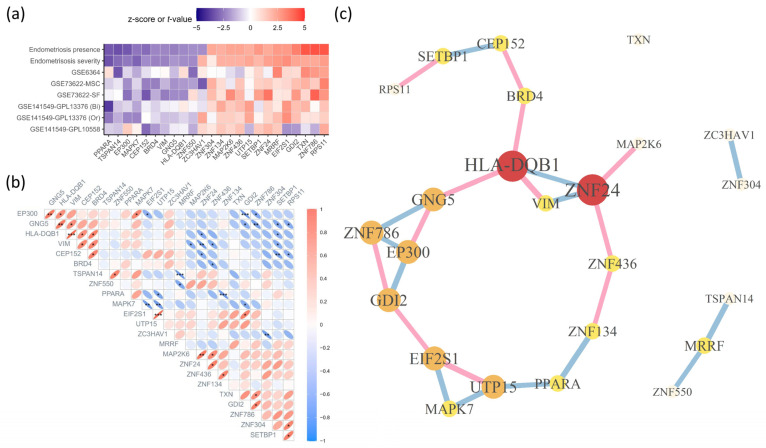
The identification of network patterns among the 24 upstream endometriosis meta genes. (**a**) The first and second rows represent the z-score calculated by IVW in METAL [10]. The z-scores in the first and second rows are the meta-analyzed combined endometriosis presence and severity signatures, respectively. The order of the genes was determined by calculating the average of two z-scores. The third to sixth rows reveal notable binomial distribution (Bi) of endometriosis presence. The seventh to eighth rows show significant ordinal distribution (Or) of endometriosis severity. (**b**) Colors, direction, density, and shape were colored differently according to Pearson’s correlation coefficient (PCC) values between the upstream meta genes. *, **, and *** indicate *p* less than 0.05, 0.01, 0.001, respectively. (**c**) The gene network is constructed based on the PCC matrix (**b**) and cut-off (*p* < 0.05). Red, orange, yellow, and khaki points indicate genes having 4 or more, 3, 2, and 1 or less of the number of interactions, respectively. The size of gene labels varies according to their significance. The color of the edges is determined by PCC, with positive values shown in pink and negative values in sky blue, while smaller *p* results in thicker edges.

## Data Availability

The data presented in this study are available in Gene Expression Omnibus (GEO) at https://www.ncbi.nlm.nih.gov/geo/ (accessed on 15 March 2024), reference numbers GSE6364, GSE73622, GSE141549. These data were derived from the following resources available in the public domain: https://www.ncbi.nlm.nih.gov/geo/query/acc.cgi?acc=gse6364 (accessed on 15 March 2024); https://www.ncbi.nlm.nih.gov/geo/query/acc.cgi?acc=gse73622 (accessed on 15 March 2024); https://www.ncbi.nlm.nih.gov/geo/query/acc.cgi?acc=gse141549 (accessed on 15 March 2024).

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
