# Peer review of "Identification of Endometriosis Pathophysiologic-Related Genes Based on Meta-Analysis and Bayesian Approach"

_ijms, 2025, doi:10.3390/ijms26010424_

Round 1

Reviewer 1 Report

Comments and Suggestions for Authors

Dear Authors

The manuscript identifies endometriosis pathophysiology-related genes using a meta-analysis based on the Bayesian approach. However, the current explanation regarding the choice and purpose of the Bayesian approach in the Introduction is not sufficiently clear, which may make it difficult for readers to follow.

Minor revision

I suggest revising the Introduction to include a more detailed explanation of:

1.      Why the Bayesian approach was chosen, highlight its advantages, such as integrating prior knowledge, handling uncertainties, and its suitability for meta-analyses where heterogeneity across datasets exists.

2.      The purpose of the Bayesian approach in this study, clarify how it improves gene identification accuracy, reduces false positives, and enhances confidence in the results compared to traditional statistical methods.

Author Response

Major comment 1. The manuscript identifies endometriosis pathophysiology-related genes using a meta-analysis based on the Bayesian approach. However, the current explanation regarding the choice and purpose of the Bayesian approach in the Introduction is not sufficiently clear, which may make it difficult for readers to follow.

Major response 1. We sincerely appreciated the reviewer’s clear critique regarding the insufficient explanation of the Bayesian approach mentioned in the title, as it highlighted an important gap in the manuscript. Gene-level studies were often hindered by characteristics that posed significant challenges to direct comparisons, such as heterogeneity among analyzed datasets, high costs, and relatively small sample sizes. To effectively address these issues, we implemented meta-analysis, a traditional method, to identify reliable endometriosis pathophysiologic-related genes. While meta-analysis selected a substantial number of genes, the Bayesian approach was employed to refine this list by integrating prior knowledge (databases and published literature) to identify key genes involved in disease pathophysiology. In this study, we utilized the Bayesian method, also known as convergence functional genomics, to narrow down hub genes related to the pathophysiology of endometriosis. To address the reviewer’s concerns, we expanded and structured the final paragraph of the introduction to provide a more comprehensive explanation of these methods and their applications.

Minor comment 1. Why the Bayesian approach was chosen, highlight its advantages, such as integrating prior knowledge, handling uncertainties, and its suitability for meta-analyses where heterogeneity across datasets exists.

Minor response 1. We were deeply grateful for the reviewer’s valuable feedback highlighting the precise advantages of the Bayesian approach that were missing from the manuscript. We fully agreed that the omission of these clearly articulated benefits, as pointed out by the reviewer, represented an area for improvement. The mentioned advantages were examined in the context of how they were applied in this study, and references supporting these applications were identified and incorporated into the first paragraph of the discussion section. Once again, we sincerely thanked the reviewer for contributing to this constructive revision.

Minor comment 2. The purpose of the Bayesian approach in this study, clarify how it improves gene identification accuracy, reduces false positives, and enhances confidence in the results compared to traditional statistical methods.

Minor response 2. We greatly valued the reviewer’s insightful feedback, which highlighted areas for improvement based on a clear understanding of the role of the Bayesian approach in this study. The Bayesian approach is indeed utilized to overcome the limitations of conventional statistical analyses by reducing false positives associated with simple p-values and enhancing accuracy. Based on this accurate critique, we clarified in the discussion section the specific objectives for employing the Bayesian approach in this research.

Reviewer 2 Report

Comments and Suggestions for Authors

The manuscript titled "Identification of Endometriosis Pathophysiologic-related Genes Based on Meta-analysis and Bayesian Approach" aims to identify genes associated with endometriosis through a combination of meta-analysis and Bayesian analysis. The study utilized transcriptomic datasets from the Gene Expression Omnibus (GEO) to perform differential expression and meta-analyses, identifying approximately 160 significant genes linked to endometriosis. Bayesian analysis further prioritized 24 genes with high-confidence scores, highlighting HLA-DQB1 and PPARA as central to the network. The study emphasizes the potential of these findings in identifying biomarkers and therapeutic targets for endometriosis. The integration of meta-analysis and Bayesian analysis provides a robust method for identifying and prioritizing genes associated with endometriosis.

Suggestions for Improvement:

1.         The historical perspectives on endometriosis (e.g., Sampson’s theory) are lengthy and not directly relevant to the study’s objectives.

2.         The inclusion criteria for datasets and genes need to be clarified.

3.         Provide explicit details on the pre-processing, normalization, and statistical methods used to ensure reproducibility.

4.         Discuss the potential limitations of the Bayesian approach, particularly its reliance on external databases with varying levels of quality.

5.         Sufficiently discuss the biological implications of findings related to ZNF family genes and their upstream role in endometriosis.

6.         Discuss the potential pathways connecting PPARA and HLA-DQB1 to endometriosis pathophysiology in greater detail.

Author Response

Comment 1. The historical perspectives on endometriosis (e.g., Sampson’s theory) are lengthy and not directly relevant to the study’s objectives.

Response 1. We greatly appreciate the reviewer’s insightful comments regarding the traditional perspectives on endometriosis. While the section may appear extensive, it was included to provide a thorough context for the study. The understanding of endometriosis has historically been dominated by traditional theories, such as Sampson’s, which have served as the foundation of its pathophysiological framework. However, recent research, including the findings presented in this study, has begun to shift the focus toward gene-level hypotheses that offer novel insights into the disease’s mechanisms and potential therapeutic targets. To underscore the significance and necessity of our research, it was deemed essential to first elaborate on these traditional perspectives before introducing the novel hypotheses explored in this work. That said, the concern about the section's length is acknowledged, and revisions have been made to streamline the discussion. The historical overview has been condensed to ensure its relevance while aligning more closely with the study’s objectives. It is hoped that these changes address the reviewer’s suggestions and contribute to greater clarity and focus in the manuscript.

Comment 2. The inclusion criteria for datasets and genes need to be clarified.

Response 2. Grateful for the effort in highlighting the omission of detailed criteria for selecting gene expression datasets from GEO, the following revisions were made. The initial search keyword was "endometriosis," and studies involving animal experiments were excluded, focusing exclusively on results derived from Homo sapiens. To ensure a comprehensive analysis of human genes, the study type was set to "Expression profiling by array." Additionally, datasets with sufficient sample sizes, including normal control groups, were ultimately selected to facilitate the analysis of endometriosis presence and severity. These inclusion criteria were added as Section 4.1 in the Methods, positioned at the beginning of the previous sections.

Comment 3. Provide explicit details on the pre-processing, normalization, and statistical methods used to ensure reproducibility.

Response 3. The lack of reproducibility was a significant weakness, and we appreciate this concern being highlighted. To address this, we referred to Reference 20 (Lee et al., Cells 2022, 11, 2867), which outlines the methodology in our previous gene expression research. However, as noted, the description provided was overly brief, potentially hindering readers’ understanding. Additional details regarding the log-transformation and distribution modifications used for normalizing gene expression values have been incorporated. Furthermore, steps involving model construction for differential expression analysis, statistical computation, and subsequent calculations for meta-analysis have been added to the methods section.

Comment 4. Discuss the potential limitations of the Bayesian approach, particularly its reliance on external databases with varying levels of quality.

Response 4. We sincerely acknowledged the reviewer’s observation regarding the potential biases stemming from the fundamental differences among the analyzed datasets. Indeed, the three gene expression datasets utilized in this study were highly heterogeneous. Consequently, simply identifying common genes after DEG analysis could have resulted in significant analytical errors. To address this issue, we identified confounders present in patient-control and severity comparisons within each dataset and incorporated these into the model construction for meta-analysis, optimizing the likelihood. A detailed explanation of this process was included in the discussion and the methods section (Section 4.1).

Comment 5. Sufficiently discuss the biological implications of findings related to ZNF family genes and their upstream role in endometriosis.

Response 5. We genuinely appreciated and valued the comment highlighting the need for a more detailed discussion on the ZNF family. As transcription factors, the ZNF family has been associated with a wide range of diseases and cellular signaling pathways. Following the reviewer’s suggestion, we conducted a literature review to investigate pathways potentially linked to endometriosis. Based on this review, we included a discussion on ZNF304, which is associated with immune cells, and ZNF436, which suppresses cell proliferation, similar to PPARA and HLA-DQB1. The relevant findings and their implications were incorporated into the discussion section.

Comment 6. Discuss the potential pathways connecting PPARA and HLA-DQB1 to endometriosis pathophysiology in greater detail.

Response 6. We fully agreed with the critique that the application of the Bayesian approach to uncover new insights from prior knowledge regarding the two most significant genes, along with a discussion on the pathways potentially involved, was a major omission. We sincerely appreciated this observation. To address this, an analysis of pathways associated with these two genes, beyond the diseases they may cause, was conducted, and a discussion on the anticipated pathophysiological conditions was added. A discussion was added suggesting that endometriosis may be associated with an abnormal immune state, based on the observation that PPARA was related to the expression of IL-6, while HLA-DQB1 appeared to be associated with CD4+ T cells.

Round 2

Reviewer 2 Report

Comments and Suggestions for Authors

The authors have addressed all of my concerns.